# E2SVM: Electricity-Efficient SLA-aware Virtual Machine Consolidation approach in cloud data centers

**Vaneet Kumar**[1], **Aleem Ali**[1], **Payal Mittal**[2], **Ibrahim Aqeel**[3]*, **Mohammed Shuaib**[3], **Shadab Alam**[3], **Mohammed Y. Aalsalem**[3]

**1** Department of Computer Science and Engineering, Chandigarh University, Gharuan, India, **2** University Institute of Pharma Sciences, Chandigarh University, Gharuan, India, **3** College of Engineering and Computer Science, Jazan University, Jazan, Saudi Arabia

* iahmed@jazanu.edu.sa

**Data Availability Statement:** All relevant data are within the manuscript.

**Funding:** The authors extend their appreciation to the Deputyship for Research& Innovation, Ministry of Education in Saudi Arabia, for funding this research work through the project number ISP-

## Abstract

Cloud data centers present a challenge to environmental sustainability because of their significant energy consumption. Additionally, performance degradation resulting from energy management solutions, such as virtual machine (VM) consolidation, impacts service level agreements (SLAs) between cloud service providers and users. Thus, to achieve a balance between efficient energy consumption and avoiding SLA violations, we propose a novel VM consolidation algorithm. Conventional algorithms result in unnecessary migrations when improperly selecting VMs. Therefore, our proposed E2SVM algorithm addresses this issue by selecting VMs with high load fluctuations and minimal resource usage from overloaded servers. These selected VMs are then placed on normally loaded servers, considering their stability index. Moreover, our approach prevents server underutilization by either applying all or no VM migrations. Simulation results demonstrate a 12.9% decrease in maximum energy consumption compared with the minimum migration time VM selection policy. In addition, a 47% reduction in SLA violations was observed when using the medium absolute deviation as the overload detection policy. Therefore, this approach holds promise for real-world data centers because it minimizes energy waste and maintains low SLA violations.

## Introduction

Cloud computing is an architecture that delivers a cost-effective, flexible, and shared pool of resources to its clients [1]. These resources can be hardware or software (e.g., memory, CPUs, bandwidth, services, and applications), and serve in a pay-as-you-go model. With this client's favorable features, cloud computing has become a popular computing platform for users [2]. Although edge computing reduces the computational load of cloud data centers (CDCs) even than emerging technologies powered by the cloud, such as blockchain, and the Internet of things (IoT), they generate heterogeneous data traffic for processing on CDCs [3, 4].

According to a report by Globe Newswire, cloud data centers are projected to experience a compound annual growth rate of 17.5%, i.e. from USD 0.37 trillion in 2020 to USD 0.83

2024. The funders had no role in study design, data collection and analysis, decision to publish, or preparation of the manuscript.

**Competing interests:** The authors have declared that no competing interests exist.

trillion by 2025 [5]. A statistical survey has shown that data centers consume 7% of the world-wide generated electricity, with an anticipated growth of 13% by 2030 [6]. Poor server utilization provokes energy consumption as an idle state server consumes approximately 70% of its peak power [7]. Another study indicates that servers typically operate at less than 50% of their maximum capacity [8]. Beloglezove et al. proposed an architectural framework for optimizing data center resources called VM consolidation.

This process is divided into four layers (1) underutilized server detection to put them in low power mode; (2) determining over-utilized hosts for smooth provisioning of requested resources; (3) selection of VMs for migration; and (4) assigning selected VMs to active or reactivated hosts [9]. Although VM consolidation maximizes resource utilization by shifting the workload from little-loaded servers and turning them to power-saving mode, it introduces a new problem of performance degradation with aggressive VM migrations that may raise SLA violations [10]. Existing studies work on a joint solution considering adaptive threshold (underload/overload detection) for maintaining low SLA violations and VM selection as well as VM placement for energy efficiency [11–13] Nevertheless, none of the existing VM selection approaches consider a reduction in SLA violations while minimizing energy consumption.

A schematic diagram of energy consumption and SLA violation tradeoff is shown in Fig 1. The hashed arrow represents reduced energy consumption by minimizing active servers and reduced SLA violations by minimizing unnecessary migrations. To stabilize the tradeoff between SLA violations and energy consumption, third and fourth-layer policies for VM consolidation are developed.

The key contributions of our research can be summarized below:

- A novel Energy-Efficient SLA-aware Virtual Machine ($E_2$SVM) consolidation algorithm is designed to deal with energy-SLA tradeoff. This algorithm significantly reduces unnecessary migrations and thereby controls the reactivation of hosts.

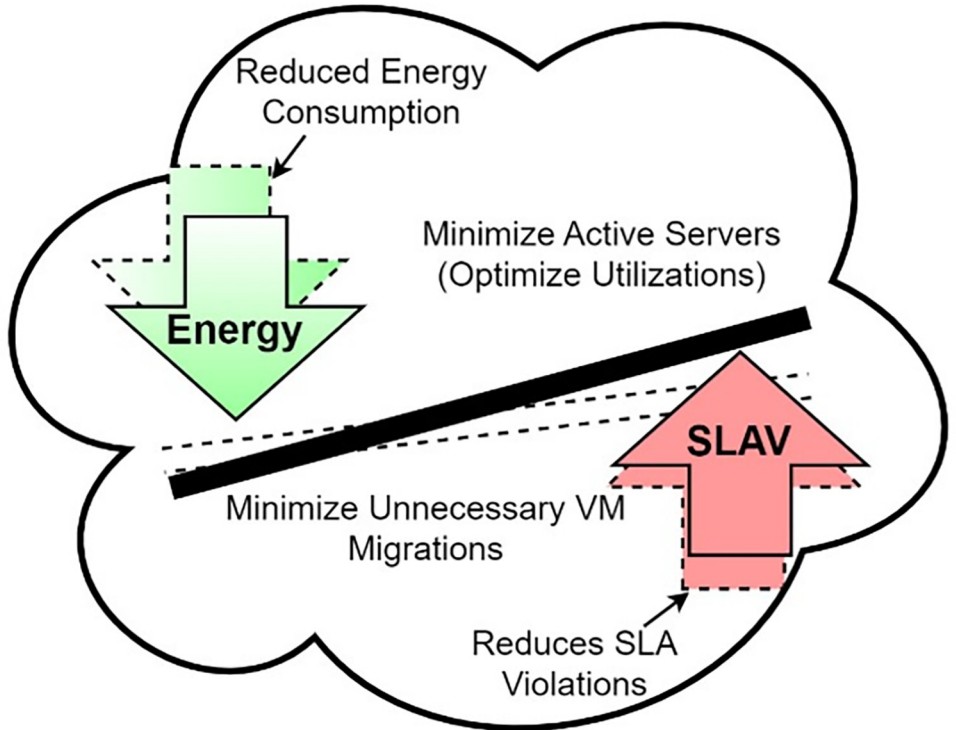

**Fig 1. The trade-off between energy consumption and SLAV in cloud data center.**

- A dynamic Minimum utilization Maximum deviation (MuMd) algorithm is proposed to select VMS from an over-utilized server. The basic idea of this algorithm is to migrate the VMs to improve the stability of the consolidation process, thereby maintaining SLA.

- We introduced the MuMd aware Best Fit Decrease (MuMdBFD) VM placement algorithm. This algorithm maximizes resource utilization with optimal use of fluctuations in resource provisioning.

The efficiency of the proposed algorithm has been studied using computer-based simulations implemented in MatLab and exhibits its superiority over existing VM selection policies. The proposed algorithm is assessed on dynamic workload to check its validity. Compared with other energy-aware algorithms the proposed algorithm can handle unnecessary VM migrations with heavy load fluctuations and shows a maximum 27.3% reduction in the number of migrations.

The paper is structured as follows: related work section provides a comprehensive review of the literature, focusing on the research challenges concerning energy-efficient SLA-aware VM consolidation. Material and methods section details the application of the E2SVM algorithm. In Experiment Setup and Efficiency Matrices section, the cloud data center and diverse metrics used to evaluate the performance of the proposed method are discussed. Results and Discussion section presents the results obtained from the proposed study. Finally, Conclusion and Future Work section outlines the conclusions of this study and future research works.

## Related work

The energy consumption is linearly proportional to CPU utilization, while the quality of Services (QoS) in terms of SLA is controlled by migration thrashing [14]. Although abundant studies have sought to address these issues, even then minimization of SLA violations, maximization of resource utilization, and energy efficiency are still challenging issues [15]. Authors [16] analyzed $EC_2$ instances and found that 20% of performance degradation is due to poor communication systems. Authors in [17] proposed adaptive-based algorithms focusing on past CPU utilization of allocated virtual machines to detect or predict overloaded hosts. Authors in [18] proposed load aware three gear threshold (LATHR) overloaded host detection algorithm to encircle the power of adaptive threshold and static threshold policies. Before these approaches, Beloglazov and Buyya proposed various adaptive and statistical methods to fix the upper threshold values required for the migration process [19]. In [20], the authors have designed multi-resource selection (MRS) VM consolidation using the concept of hottest resource to determine overloaded hosts. They introduced the concept of resource temperature and resourced correlation to determine potential VM for migration.

After detecting an overloaded host, VM selection has always been a part of VM consolidation to make an optimal decision. Beloglazov and Buyya proposed minimum migration time (MMT), maximum correlation (MC), and random selection (RS) VM selection policies. The RC policy selects VMs randomly to reduce the load from the host. The MMT approach selects the VM with least memory utilization among all the VMs from the overloaded host, taking minimum migration overhead. The MC approach selects VM with maximum correlation and proves more power saving but lacks QoS. In contrast, MMT provides better performance incurring more power.

Additionally, Zhou et al. [21] offered five VM selection rules and a three-threshold energy-saving algorithm (TESA) to enhance SLA. Furthermore, Zhou et al.'s [22] innovative flexible three-threshold energy aware (ATEA) VM placement method has decreased energy use and SLA violations. Minimum memory size and lowest CPU utilization constraints are taken for

designing VM selection policies. Monil and Rahman have introduced fuzzy logic-based energy-saving VM consolidation algorithms [23]. The standard deviation, memory size, and correlation have been used for defuzzification to select VMs for migration.

Further, Yadav et al. [24] have come up with an idea of 'M estimation regression' (MeReg) for managing the energy-SLA tradeoff. They used VM's CPU utilization history to detect the hotspot server to address this issue. In continuation, authors in [25] have proposed two adaptive heuristic models, gradient recent-based regression (Gdr) and bandwidth responsive algorithm for VM selection whereas maximum correlation percentage (MCP) to fix dynamic upper threshold. Noteworthy progress has been recounted regarding energy efficiency and QoS in the cloud data center (DC). Yadav et al. [26] have extended the research for over-utilized host detection using the least medial square regression (LmsReg) heuristics algorithm and introduced a utilization-based prediction method for VM selection. These heuristic algorithms significantly reduced energy consumption with improved SLA violations. This approach is high energy saving but lacks in maintaining low SLA violations. In the latest work by Mandal et al. [27], they introduced an approach aimed at relocating VMs with a dual objective: minimizing size while consuming high energy. While this approach demonstrates significant energy savings, it falls short in maintaining low SLA violations.

All the above approaches improved energy efficiency, focusing on unnecessary VM migrations by simultaneously overlooking fluctuations and current CPU utilization. Migrations of VMs with constant load can fall the DC in migration thrashing (MT) due to redundant migrations resulting in performance degradation. Similarly, heavily loaded VMs should not be migrated to achieve minimum live migration cost. As studied in the literature that requested workload is uncertain in the real world. Thus, utilization aware robust statistical method needs to be designed for handling dynamic consolidation.

## Materials and methods

In this section, we will discuss the resource provisioning architecture of VM consolidation in a dynamic environment, which involves using virtual machines as shown in Fig 2 [28]. The main challenges of this architecture are achieving maximum resource utilization and delivering high-quality services. The quality of service is measured using SLA metrics. In order to optimize resource utilization, VMs with high remaining utilization and maximum deviation are migrated to hosts with high remaining utilization and minimum deviation [29].

The fundamental process of consolidating VMs to enhance resource utilization and guarantee QoS maintenance is shown in Algorithm 1.

ALGORITHM 1: ENERGY-EFFICIENT SLA AWARE VIRTUAL MACHINE (E2SVM) CONSOLIDATION ALGORITHM

**Input:** $h_{list}$, $vm_{list}$
**Output:** $h_{chosen}$: Detected destination host
1. for each host h in data center
2. for each VM on current host
3. Step-1: $Uthr_h$ = MeanAbsoluteDeviation ($h_{list}$)||DoubleThreshold
4. Step-2: $MigVM_{list}$ = MuMd ($Host_{over}$, $Uthr_h$)
5. Step-3: $h_{chosen}$ = MuMdBFD ($H_{list}$, $MigVM_{list}$, $uth$)
6. end

The process begins with selecting the list of allocated VMs as input. Then, the workload on each VM is determined at regular time intervals. During each interval, overutilized and underutilized hosts are identified using MAD and THR policies. In the subsequent steps, we delve into the discussion of a dynamic workload-aware VM selection policy called Minimum Utilization Maximum deviation (MuMd) under subsection A. By migrating VMs with minimal utilization, energy efficiency can be enhanced. Additionally, by considering maximum deviation, we can control fluctuations in host status, thereby reducing the number of migrations. This

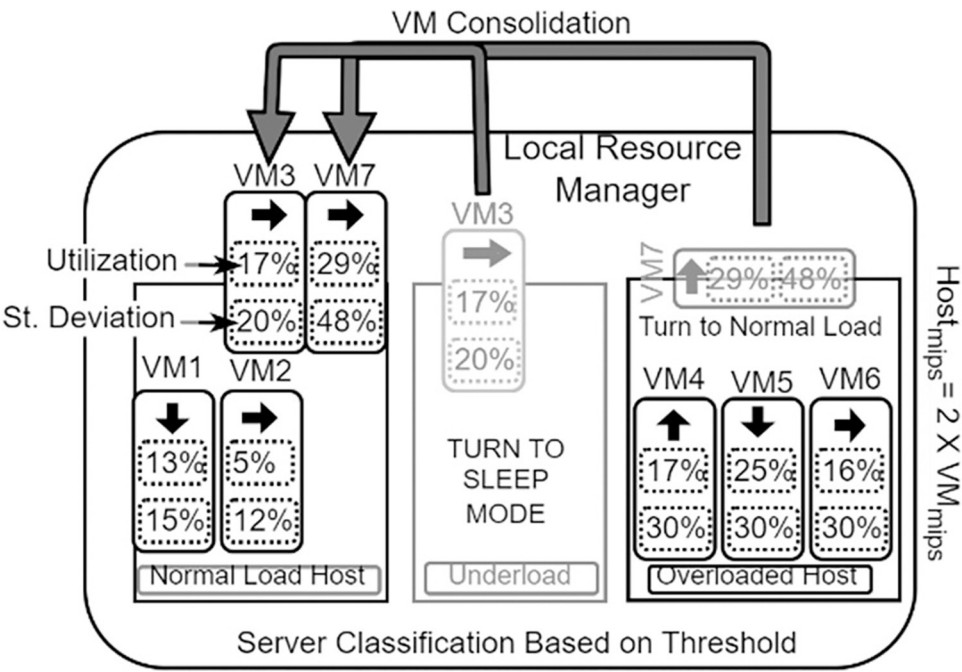

**Fig 2. Multi-objective VM consolidation architecture.**

approach is important because VMs with high standard deviation and low current utilization on a host are more likely to cause host overload once again. Following this, the selected VMs from the previous step are mapped to suitable hosts utilizing the suggested VM placement algorithm discussed in subsection B. The combined behavior of these proposed algorithms promotes higher resource utilization, controlled migrations, and ultimately leads to improved SLA compliance and energy efficiency. Table 1 provides a list of notations used throughout this paper.

## Proposed MuMd based VM selection algorithm

It has been proved that prudent migrations can optimize VM consolidation and save unwanted traffic load. Therefore, SLA-obedient energy-efficient VM selection is as important as overutilized and underutilized host detection. The suggested algorithm calculates the CPU fluctuations ratio with the summation of current CPU utilization and fluctuations individually for all the virtual machines using the following equation [30]. Further, one or more VMs with the maximum ratio is being selected for migration, explained using flowchart shown in the

**Table 1. Notations.**

| | |
|---|---|
| $V_h$ | Number of virtual machines |
| $\delta_v$ | Deviation in utilization of VM *for* last n iterations |
| $L_v$ | Load ratio of current VM |
| $curr\_uti_v$ | Current utilization of VM $v$ |
| $mips$ | Million instructions per second |
| $CPU_{mips}(v_\tau)$ | CPU utilization in *mips* at time $\tau$ |
| $aveCPU_{mips}(v)$ | Average CPU utilization in *mips* of VM $v$ |

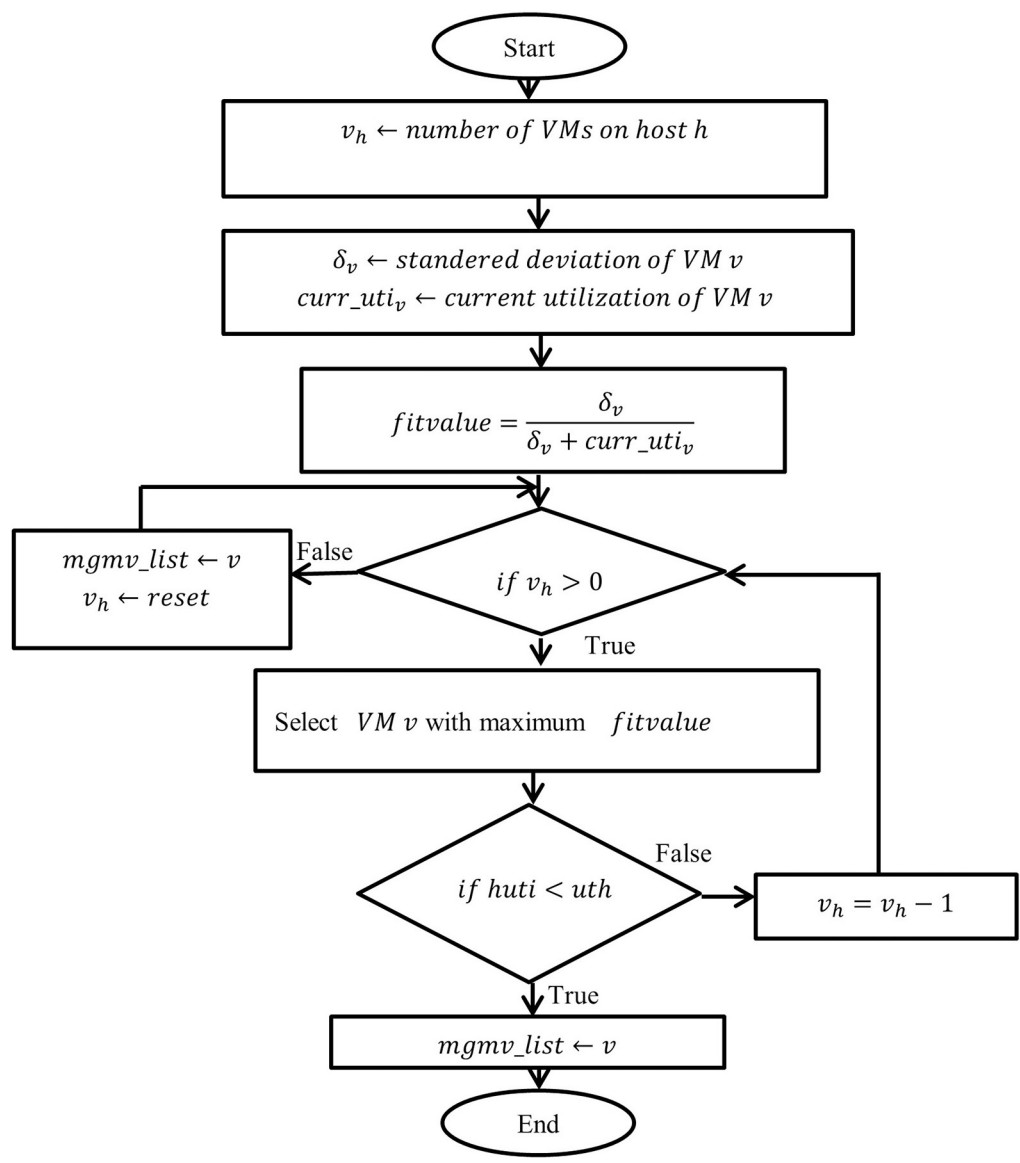

**Fig 3. Flow chart of proposed technique (MuMd).**

Fig 3.

$$V_h = \{v | v \text{ is a VM allocated on the host } h\}$$

$$ratio = \frac{\delta_v}{\delta_v + curr\_uti_v} \times L_v \tag{1}$$

$$L_v = \frac{curr\_uti_v}{availability} \tag{2}$$

Where the set of allocated VMs on host h are represented by $V_h$. The current CPU utilization $curr\_uti_v$ of virtual machine $v$ is unitized according to standard deviation ($\delta_v$) for last n iterations as referred in (3).

Let us consider $CPU_{mips}(v_\tau)$ representing the MIPS allocated to virtual machine $v$ at time $\tau$. The CPU utilizations of previous $itr$ number of different time slots are denoted as $CPU_{mips}(v_{\tau-1})$, $CPU_{mips}(v_{\tau-2}) \ldots\ldots\ldots CPU_{mips}(v_{\tau-itr})$ and $aveCPU_{mips}(v)$ is the average CPU utilization of virtual machine $v$.

$$\delta_v = \sqrt[2]{\frac{1}{itr}\sum_{i=1}^{itr}(CPU_{mips}(v) - avgCPUmips(v))} \tag{3}$$

Algorithm 1 shows the implementation of the proposed work in two phases. In the first phase, Line 5- Line 8 determine the CPU utilization ($huti$) of host in MIPS. As indicated by Line 10 onwards, in the second phase, one or more VMs are selected until the victim host changes its state to normal load, narrated in Fig 2. Line 13 is used to find the ratio of CPU utilization of individual VMs as represented in (1). The best virtual machine will be selected as indicated by Line 14–16 after finding the ratio of all allocated VMs. The proposed algorithm (MuMd) is represented in algorithm 1, and Fig 3 represented the flow of algorithm pictorially.

ALGORITHM 2 PROPOSED ALGORITHM (MUMD VMs SELECTION)

```
1.  Input: VM_list from overloaded host h
2.  Output: Selected VMs
3.  MigVM_list←NULL
4.  Huti←0
5.  MaxRatio←MIN
6.  Vh←Each and every virtual Machines on h
7.  for each v in Vh do
8.  huti←huti+v.utilization
9.  end for
10. while (huti≥Uthr)
11. for each VM v in Vh do
12. δv←Stdev(v)
13. ratio = δv / (δv+curr_utiv)
14. if ratio≥MaxRatio then
15. MaxRatio←ratio
16. vchosen←v
17. end if
18. end for
19. huti←huti-CPUmips(v)
20. MigVM_list←vchose
21. end while
22. end
```

The entanglement of the suggested algorithm depends upon the total number of hosts (M) and virtual machines (N), or O(M+N).

## Proposed MuMd based VM placement algorithm

VM placement is considered to be a bin packing optimization problem [31]. As bin packing is an NP-hard problem thus, placement for increasing idle hosts while considering intra VM dependency is a big challenge [32]. Power-aware best-fit decrease (PABFD) is the single prime objective VM placement algorithm. The flow of the proposed MuMd based Best Fit Decrease (MuMdBFD) VM Placement policy is discussed as pseudo-code in algorithm 2. The algorithm reinforces the full flow of VM allocations to the suitable servers and depicts that the system restricts SLA violations with controlled migration thrashing, hence reducing energy consumption.

ALGORITHM 3: MUMD BASED BEST FIT DECREASE ALGORITHM (MUMDBFD)

```
Input: Hlist: all hosts, vm: a new virtual machine, uth: upper
threshold
```

```
Output: h_chosen: the selected destination host
 1. h_chosen ← null
 2. hratio ← 0
 3. MinRatio ← MAX
 4. Choose cluster c
 5. for each host h in HC_c do
 6. huti ← 0
 7. Vh ← All virtual Machines on h
 8. for each v in V_h do
 9. huti ← huti+v.utilization
10. hratio = hratio + δ_v/(δ_v+curr_uti_v)
11. end for
12. if huti>0 then // check active/inactive state of host
13. utilization ← huti+v.utilization
14. if utilization>1 then
15. continue
16. else if utilization≤uth then
17. ratio = hratio + δ_vm/(δ_vm+curr_uti_vm)
18. if ratio ≤ MinRatio then
19. MinRatio ← ratio
20. h_chosen ← h
21. endif
22. endfor
23. Return: h_chosen
```

In Algorithm 2, Line 8, CPU utilization (*huti*) of the host is determined. Line 9 determines the CPU utilization to standard deviation ratio (hratio) for each virtual machine (VM) allocated to host h. The purpose of obtaining *hratio* is to identify a suitable host where the unsettled CPU utilization is maximized for VM placement. Overall deviation is minimum among all the hosts as described by line 16-line 18. After all iterations algorithm returns the best host with high remaining utilization and comparatively minimum deviation.

*Theorem 1*: The time complexity of the algorithm $O(M \times N)$ and depends upon the total number of hosts (M) and virtual machines (N).

*Proof*: $E_2SVM$ algorithm comprises of three steps in sequence. Initialy, adaptive threshold is determind for overload server detection in $O(M \times k)$ time, where k is the number of virtual macines on a server. In the second step VM is selected in $O(M+N)$ time. Finally, VMs in migration list are allocated or placed in suitable host in $O(M \times N)$ time. Therefore, the overall time complexity of the proposed algorithm is $O(M \times N)$.

## Experiment setup and efficiency matrices

In this section, we will explain the simulation setup of a cloud data center for comparison and evaluation of existing algorithms with proposed VM selection approach. Thereafter, basic definitions (includes energy consumption models, energy efficiency metric, and SLA violation metrics). In the end, results and discussion are presented in the next section. Table 2 below provides list of abbreviation and notations used to define the different efficiency matrices used in this study.

## Modeling of physical machines and VM requests

The experimental setup to run application on IaaS environment, involves configuring N virtual machines $\{V = (v_1,v_2,\cdots,v_N)\}$ and M servers $\{S = (s_1,s_2,\cdots,s_M)\}$. Each host *h* is delineated by its processing capacity ($CPU_{mips}(h)$) in MIPs and $PC_h$ as its power usage is measured in watts per second. Likewise, the processing capacity of each virtual machine, represented by $v \in (v_1, v_2,\cdots,v_N)$. is characterized by $CPU_{mips}(v)$.we simulate a data center with 290 virtual machines

**Table 2. Abbreviations and notations.**

| $H_h$ | Set of $M$ hosts |
|---|---|
| $CPU_{mips}(h)$ | CPU utilization of host $h$ in *mips* |
| $CPU_{mips}(v)$ | CPU utilization of VM $v$ in *mips* |
| $PC_h$ | Power consumption of host $h$ |
| $PC^{max}$ | Power consumption at full utilization |
| $CPU_{mips}(h_\tau)$ | CPU utilization of host in *mips* at time $\tau$ |
| $EC_h$ | Energy consumption by host $h$ |
| $TEC$ | Total energy consumed by data center |
| $a_{vh}$ | Allocation of $v$ on host $h$. |

and 100 physical hosts of heterogeneous configuration as mentioned in Tables 4 and 5.

## Modeling of energy consumption

The CPU utilization, measured in MIPS, is recognized as a crucial factor in governing energy consumption because the power consumption ($PC_h$) of a server is directly proportional to its CPU usage, as indicated in Eq 4 [33]. To accurately describe the energy consumption, we use real data on energy consumption, all of which derive from the SPECpower benchmark [34]. For the experimental study of the proposed work, the HP G4 server is utilized and details of energy consumption is illustrated in Table 3. The maximum power ($PC^{max}$) consumed by the HP G4 server amounts to 117 watts [35]. It is observed that, when a server remains idle, it still consumes approximately 70% of its total power. Consequently, underutilized servers consume more power per instruction (about 0.7 times) compared to fully utilized servers.

$$\text{C. } PC_h = \begin{cases} 0.7 \times PC^{max} \times CPU_{mips}(h) + 0.3 \times PC^{max}, & CPU_{mips}(h) > 0 \\ 0, & Otherwise \end{cases} \tag{4}$$

Due to the dynamic nature of workloads, the CPU workload of a host ($h$) differs over time ($\tau$), which can be represented as $CPU_{mips}(h_\tau)$. As a result, the energy usage ($EC_h$) over a given time period can be considered an inherent factor of power consumption.

$$\text{D. } EC_h = \int_{\tau_0}^{\tau_1} PC_h(CPUmips(h_\tau))dt \tag{5}$$

## SLA Metric

Beloglazov and Buyya provided an explanation for SLA violations within the context of the Infrastructure as a Service (IaaS) model, as demonstrated in Eq (6). They introduced the concept of SLATH, which represents the time duration during which SLA violations occur on an active host operating at maximum CPU utilization. Additionally, they introduced PDM as an abbreviation for Performance Degradation by VM resulting from live migrations.

$$SLAV = SLATH \times PDM \tag{6}$$

**Table 3. Power consumption by the two servers at different load levels in watts.**

| Server Utilization | 0% | 20% | 40% | 60% | 80% | 100% |
|---|---|---|---|---|---|---|
| Power Consumption | 86 | 92.6 | 99.5 | 106 | 112 | 117 |

## IER Metric

Instruction Energy Ratio is another metric that consider both the performance and energy consumption in Million instructions per joule as referred in (7).

$$IER(\mathcal{M}) = \frac{\sum_{h=1}^{M} \sum_{v=i}^{N} (a_{vh}.CPUmips(v))}{TEC} \tag{7}$$

This study's primary objectives are to increase the number of instructions carried out per unit of energy usage and reduce SLA breaches.

## Results and discussion

The effectiveness of the proposed algorithm is evaluated through computer-based simulation conducted in Matlab version R2017a. The specifications of the hosts and virtual machines, obtained from prior studies, are presented in Tables 4 and 5. To ensure result accuracy, the data center undergoes ten iterations during the simulation. The validation of the findings is done by examining SLA violations and energy efficiency metrics. Additionally, the previous section on Experiment Setup and Efficiency Matrices discusses the utilization of IER to demonstrate energy savings per instruction

The effectiveness of the proposed algorithm is evaluated against existing VM selection policies, namely minimum migration time (Mmt), maximum correlation (Mc) [36], and maximum utilization minimum size (MuMs) [37]. In the Mmt policy, the VM with the least remaining time to complete migration is chosen. Conversely, in the Mc policy, the VM with the highest correlation among others on the oversubscribed server is selected for migration to alleviate the workload below the threshold level. In our simulations, we utilize the medium absolute deviation (MAD) and interquartile range (Iqr) policies to establish threshold levels.

The proposed approach reduces energy consumption by an average of 9.9% and 9% compared to the Mad and Iqr algorithms, respectively, as shown in Figs 4 and 5. Fig 6 depicts the energy consumption in joules per million instructions using IER. According to the interpretation, the proposed approach demonstrates a significant 13.3% improvement in the instructions performed per unit of energy.

SLA violations are a critical concern for cloud service providers because financial penalties may be imposed by users for the duration of the performance degradation. SLA is considered to be a standard parameter to determine the quality of service. The proposed approach in the present study reduces an average 10% SLA violations as shown in Fig 7.

After selecting the victim VMs, it needs to be migrated to other suitable host machines. Thus, migration of virtual machines also degrades the desired performance level and imparts processing overhead to the entire system. Fig 8, reveals that the suggested algorithm has a significantly lower number of migrations than MM and RC algorithms. In order to place

**Table 4. Specifications of data center.**

| Number of Hosts | Number of Virtual Machines | Data Center |
|---|---|---|
| 290 | 100 | 1 |

**Table 5. Specifications of machines.**

| Type | RAM (MB) | CPU | MIPS |
|---|---|---|---|
| Host | 8192 | 1 | 1000–3000 |
| Virtual Machine | 128 | 1 | 250–1000 |

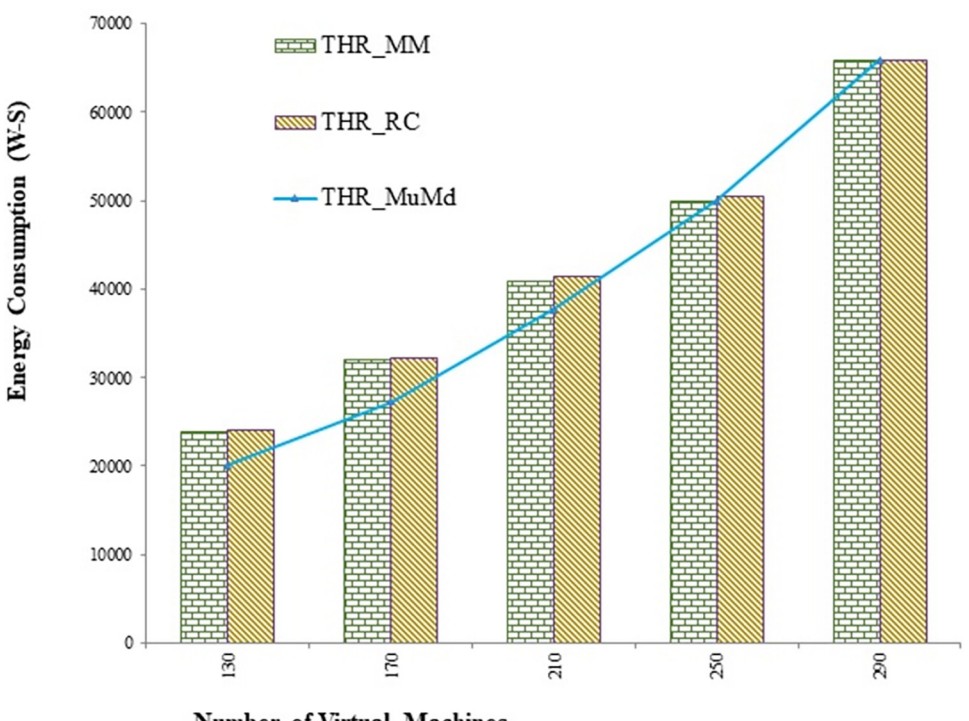

**Fig 4. Energy usage with varying VMs for double threshold (THR) as overload detection policy.**

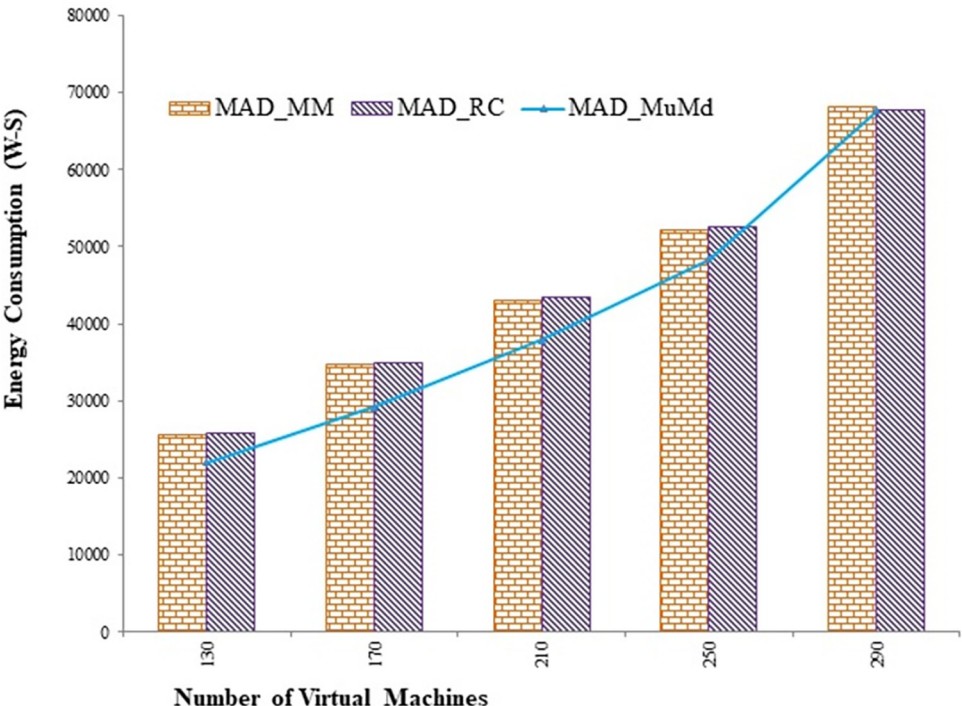

**Fig 5. Energy usage with varying VMs for Median Absolute Deviation (MAD) as overload detection policy.**

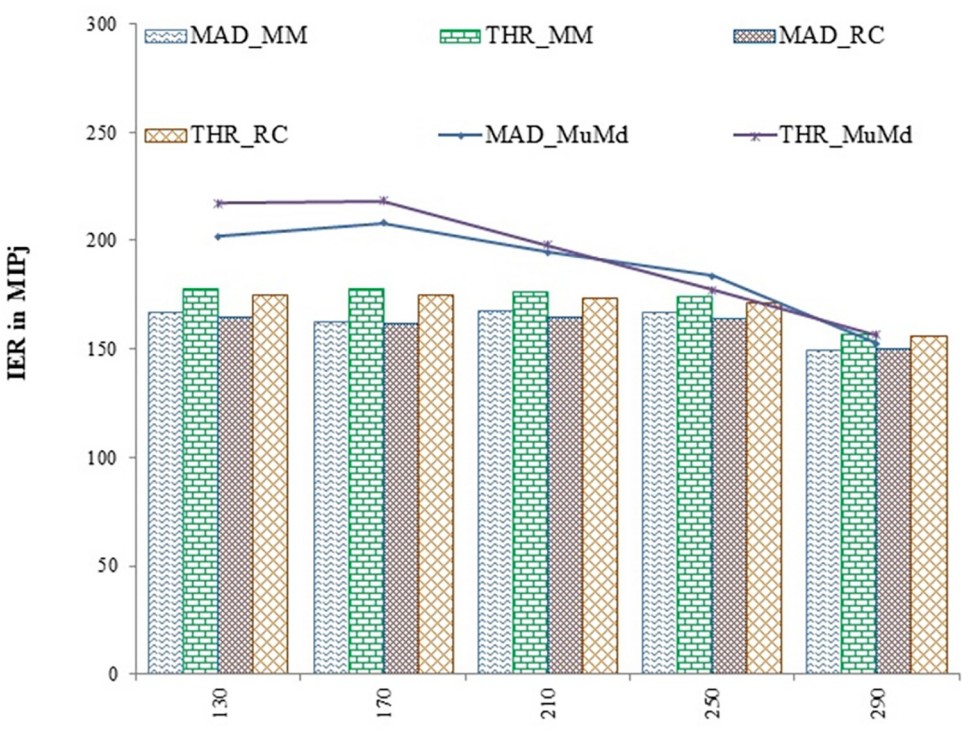

**Fig 6. Change in IER for MAD and THR as overload detection policies v/s increasing virtual machines.**

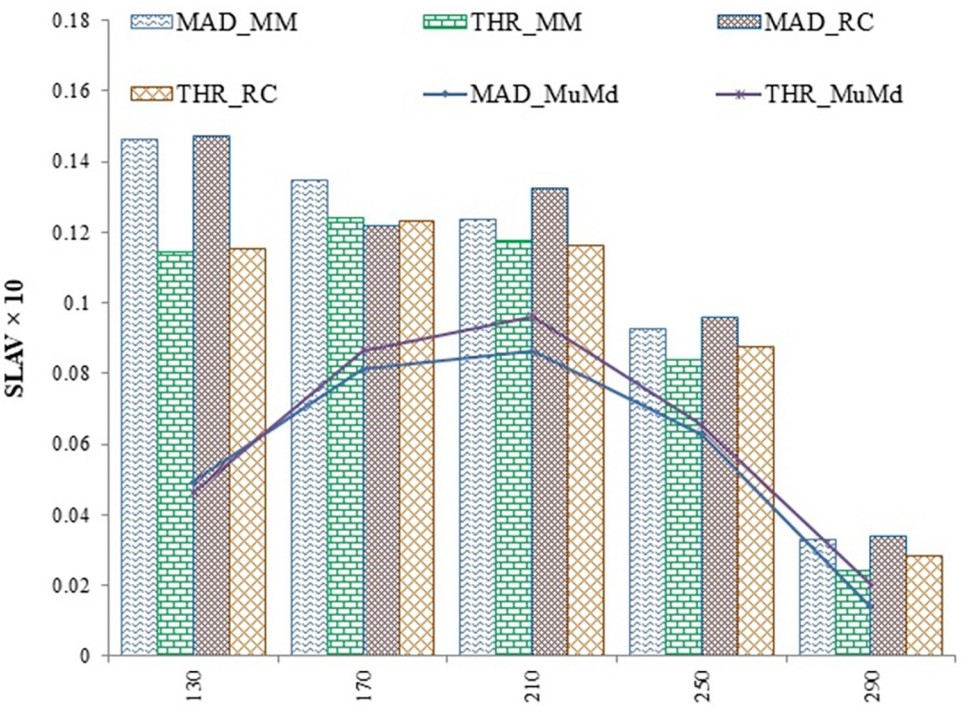

**Fig 7. SLA Violations for MAD and THR as overload detection policy v/s virtual machines.**

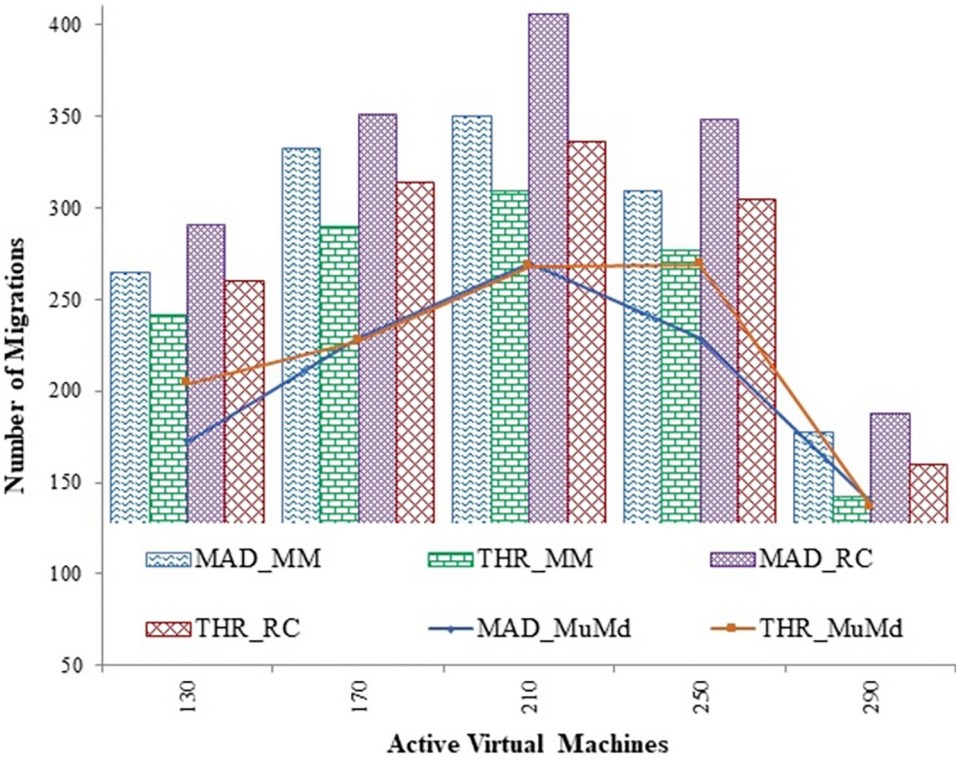

**Fig 8. Change in migrations count for MAD and THR as overload detection policies against increasing virtual machines.**

underloaded hosts in a low-power mode, all virtual machines (VMs) residing on those hosts are migrated. After the installation of migrable VMs, Fig 9 compares the number of idle hosts that are still present. The results demonstrate that the proposed technique exhibits superior performance compared to the MAD and THR algorithms, with reductions of 10.4 percent and 6.2 percent in idle hosts, respectively.

Results in Fig 10 reveals the improvement of overall percentage of suggested algorithm (MuMd) with Minimum Migration (MM) and Random Choice (RC) VM selection algorithms using Median Absolute Deviation (MAD) and Double Threshold (THR) as host overload detection policies. It can be clearly noted that the proposed scheme offers improvement for energy consumption, idle servers, migrations count, instruction energy ratio and service level agreement.

## Conclusion and future work

In this paper, we proposed an Energy Efficient SLA-aware Virtual Machine Consolidation (E2SVM) algorithm. Initially, we employed existing Median Absolute Deviation (MAD) and Interquartile Range (IQR) policies for detecting overloaded servers. Subsequently, victim virtual machines were selected based on fluctuations using the last 10 historical utilizations and current resource requirements. The proposed Minimum Utilization Maximum Deviation (MuMd) approach considered both past CPU utilization and current resource requirements simultaneously.

This study addressed two critical challenges in the IaaS cloud platform: reducing energy consumption and preventing server overutilization to minimize SLA violations. We

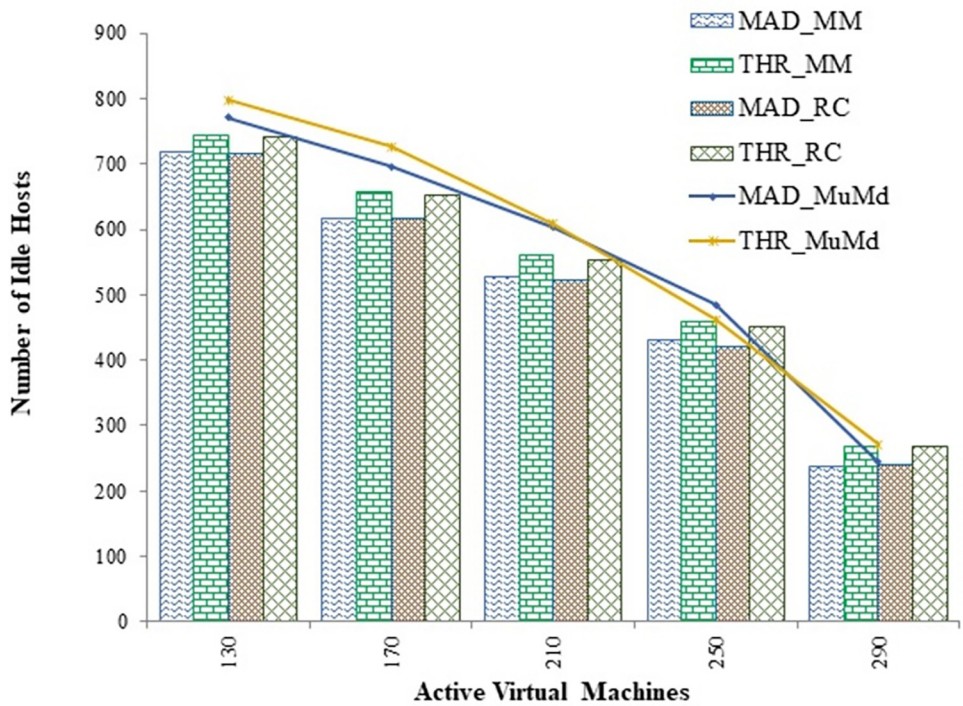

**Fig 9. Controlling idle servers for MAD and THR as overload detection policies against increasing virtual machines.**

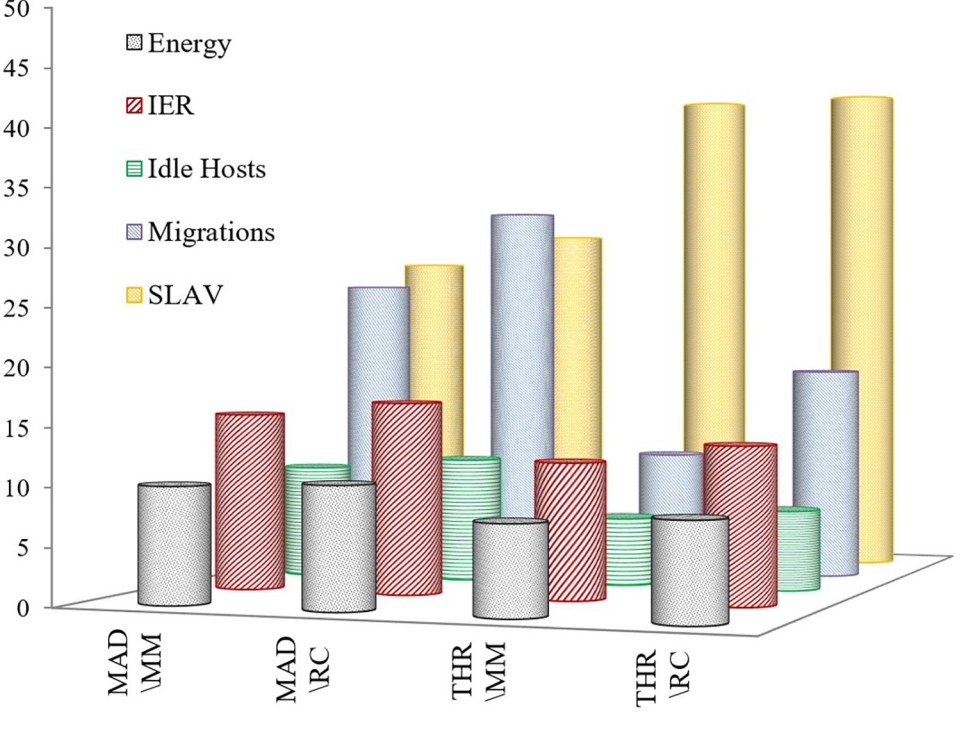

**Fig 10. Percentage improvement in various metrices using proposed algorithm compared with existing policies.**

implemented the proposed approach using computer-based simulations and evaluated it in Matlab version 2017a. The simulation results demonstrated significant improvements compared to existing benchmark algorithms, including an average reduction of 18.9% in the number of migrations, 12.7% in idle hosts, and a 9.5% decrease in electrical energy usage.

In the future, we aim to incorporate machine learning algorithms for forecasting server workload and apply our approach to real-world data centers.

## Author Contributions

**Conceptualization:** Vaneet Kumar, Shadab Alam.

**Data curation:** Mohammed Shuaib.

**Formal analysis:** Aleem Ali, Payal Mittal, Ibrahim Aqeel, Shadab Alam, Mohammed Y. Aalsalem.

**Funding acquisition:** Ibrahim Aqeel, Mohammed Y. Aalsalem.

**Investigation:** Payal Mittal, Mohammed Shuaib.

**Methodology:** Vaneet Kumar, Aleem Ali, Payal Mittal.

**Project administration:** Shadab Alam, Mohammed Y. Aalsalem.

**Supervision:** Mohammed Y. Aalsalem.

**Validation:** Mohammed Shuaib.

**Visualization:** Ibrahim Aqeel.

**Writing – original draft:** Vaneet Kumar, Aleem Ali, Shadab Alam.

**Writing – review & editing:** Payal Mittal, Ibrahim Aqeel, Mohammed Shuaib, Mohammed Y. Aalsalem.

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
