## [Decision Letter · Decision Letter 0]

27 Feb 2024

PONE-D-24-01820E2SVM: Electricity-Efficient SLA-aware Virtual Machine Consolidation Approach in Cloud Data CentersPLOS ONE

Dear Dr. Aqeel,

Thank you for submitting your manuscript to PLOS ONE. After careful consideration, we feel that it has merit but does not fully meet PLOS ONE’s publication criteria as it currently stands. Therefore, we invite you to submit a revised version of the manuscript that addresses the points raised during the review process.

We look forward to receiving your revised manuscript.

Kind regards,

Jacopo Soldani

Academic Editor

PLOS ONE

“The authors extend their appreciation to the Deputyship for Research& Innovation, Ministry of Education in Saudi Arabia, for funding this research work through the project number ISP-2024.”

“The authors extend their appreciation to the Deputyship for Research& Innovation, Ministry of Education in Saudi Arabia, for funding this research work through the project number ISP-2024.”

“The authors extend their appreciation to the Deputyship for Research& Innovation, Ministry of Education in Saudi Arabia, for funding this research work through the project number ISP-2024.”

Reviewers' comments:

Reviewer's Responses to Questions

**Comments to the Author**

1. Is the manuscript technically sound, and do the data support the conclusions?

Reviewer #1: Yes

Reviewer #2: Yes

Reviewer #3: Partly

2. Has the statistical analysis been performed appropriately and rigorously? 

Reviewer #1: Yes

Reviewer #2: Yes

Reviewer #3: I Don't Know

3. Have the authors made all data underlying the findings in their manuscript fully available?

Reviewer #1: Yes

Reviewer #2: No

Reviewer #3: Yes

4. Is the manuscript presented in an intelligible fashion and written in standard English?

Reviewer #1: Yes

Reviewer #2: Yes

Reviewer #3: Yes

5. Review Comments to the Author

Reviewer #1: To reduce energy consumption and SLA violations, the authors propose a VM consolidation algorithm. The experimental results illustrate that the proposed algorithm performs better than other algorithms in terms of energy consumption and SLA violations. However, the following problem should be addressed:

(1)The Abstract of this paper should be rewritten.

(2) In the paper, what is the time complexity for the proposed algorithm? It is better for the authors to give more explanations of the algorithms, such as the function of each step, and the overall basic idea.

(3)The authors make a comparison with other algorithms in the paper. However, these algorithms are out of date. It is not fair to make a comparison with these algorithms. Could the authors make a comparison with the newly published algorithms in the field of VM deployment, e.g., “An Energy-efficient VM Allocation Algorithm for IoT Applications in a Cloud Data Center”, 2021 (The source code and videos are shared on GitHub https://github.com/mshojafar/sourcecodes/tree/master/Zhou2021AFED-EF_Sourcecode); and “Minimizing SLA Violation and Power Consumption in Cloud Data Centers Using Adaptive Energy-aware Algorithms”.

(4)For the comparison algorithms, It is suggested to make a simple introduction in the experiment part.

(5) The quality of the Figures in the paper should be further improved in the revision.

(6) In the paper, the authors evaluate the energy consumption. However, any power model is essential for energy-aware algorithms. Have the authors considered selecting other energy consumption models? The author can benefit from these references to enrich these parts, such as “IECL: An Intelligence Energy Consumption Model for Cloud Manufacturing”, DOI: 10. 1109/TII. 2022.3165085; doi:10.1109/TGCN.2021.3121961; DOI: 10.1007/s00521-019-04119-7.

(7) There are many spelling errors in the manuscript. A thorough spelling check is required in the revision.

Reviewer #2: This paper proposes an approach to improve Virtual Machines (VMs) consolidation in a data center by balancing Service Level Agreement (SLA) violations and energy consumption. The approach is based on a multi-objective consolidation architecture that individuates under-utilised servers to be put in low power mode and over-utilised servers to migrate VMs in order to avoid performance degradation which could lead to SLA violations.

The devised algorithms are evaluated by simulation and compared with existing VM migration policies.

The topics of the paper are interesting and with growing importance in the last few years. The paper has a standard structure and, generally, is well-written with only some typos and small spelling mistakes.

Given that energy consumption is a central topic of the paper, I think the energy model should be motivated and explained better to convince the reader that energy consumption is linearly proportional to CPU utilisation. The cited energy model refers to a study from 2007 and I believe that data center technologies are changed in the meantime. Moreover, my intuition is that such linear relation is referred to CPU-bound applications only, and I do not know if can be considered for the general case. I suggest studying and discussing these points in the paper.

Moreover, I think that the used algorithm's execution should be evaluated in terms of energy consumption since the result of about 9% energy consumption in the evaluation section could be decreased by the algorithms. E.g. if the algorithms execution consumes 10% of the overall energy there is no gain in terms of energy consumption.

Furthermore, I suggest evaluating if unutilised servers can be automatically shut down to reduce energy consumption and if the power-up when they are needed is feasible in terms of time to avoid SLA violations.

In conclusion, I think that the paper can be improved but could be accepted after revision of some parts.

Reviewer #3: In this paper, the authors proposed an Energy-Efficient Service Level Agreement (SLA)-‎aware Virtual Machine (E2SVM) to control energy consumption through optimization of ‎existing resources. Overall, the paper is well organized and written in a rigorous and clear ‎style. The current article necessitates a comprehensive structural revision to enhance its ‎coherence. Please revise the following issues:‎

‎-‎ Although the ABSTRACT structure is good, I suggest that the philosophy of using the ‎proposed method should be explained.‎

‎-‎ In my opinion, the INTRODUCTION section needs to be revised. In this section there ‎should be three points: 1) motivation, 2) a summary of the challenges of previous ‎studies, and 3) contribution.‎

‎-‎ The Related Work section is too short, should have additional resources, and may be ‎separated by subjects: QoS, Energy consumption, and SLA, subjects related to the ‎article. Here are some suggestions: " Server ‎Consolidation Algorithms for Cloud ‎Computing: Taxonomies and Systematic Analysis of ‎Literature", “MECpVmS: an SLA ‎aware energy-efficient virtual machine selection policy for green cloud computing”, " ‎Reducing Energy Footprint in Cloud Computing: A Study on the Impact of ‎Clustering ‎Techniques and Scheduling Algorithms for Scientific Workflows", “Power and thermal-‎aware virtual machine scheduling optimization in cloud data center”, "Computing ‎‎Resources Scalability Performance Analysis in Cloud Computing Data Center", ‎‎"HEPGA: a ‎new effective hybrid ‎algorithm for scientific workflow scheduling in cloud ‎computing ‎environment", "Modeling and analysis of quality of service and energy ‎consumption in cloud environment”. ‎

‎-‎ The authors stated in the abstract “So, this approach can be used in real-world data ‎centers through minimizing of energy wastage thereby maintaining low SLA violations ‎or fluctuations”, but no real implementation was made in the paper. While the ‎experiment and results presented are intriguing and indicative of promise, it is ‎imperative to validate these outcomes through real-world application. I recommend the ‎incorporation of empirical experiments to facilitate a meaningful comparison with ‎simulation results; it is better to analyze the time complexity of the proposed method in ‎the worst case.‎

‎-‎ While the article alludes to cost reduction associated with Cloud services, a ‎comprehensive examination of this aspect remains elusive. It would be prudent for the ‎author to expound upon the topic, particularly focusing on unreliable services like ‎preemptible servers (e.g., Spot Instances on Amazon and Preemptible instances on ‎Google). Noteworthy articles like "MULTS: A Multi-cloud Fault-tolerant Architecture to ‎Manage Transient Servers in Cloud Computing" (Journal of Systems Architecture, 2019) ‎and "Portfolio-driven Resource Management for Transient Cloud Servers" (ACM on ‎Measurement and Analysis of Computing Systems 2019) are worth considering. The ‎viability of unreliable services could be assessed within the framework of Fault ‎Tolerance approaches, thereby underlining the potential trade-offs in terms of Quality of ‎Service (QoS). In light of these considerations, the article's focus may be refined to ‎address these intricate aspects;‎

‎-‎ The main limitation of the proposed methods should be explained in detail;‎ ‎

‎-‎ The contribution and novelty of this work should be further justified and future studies ‎‎should be given;‎ ‎

‎-‎ The simulation environment should be provided in detail, e.g., software, sampling size, ‎solver ‎type, etc.‎

‎-‎ An important point needs to be included in this article: the type of application that can ‎be executed using the proposed model. The inclusion of this facet would enrich the ‎article's scope and applicability;‎

‎-‎ There are still some grammatical errors in the manuscript. Authors should use software ‎such as Grammarly for proof-checking. ‎

‎-‎ The tense of the verbs in the CONCLUSION section must be past tense. In this section, ‎the most important numerical improvements of the proposed method should be ‎mentioned and marginal explanations should be avoided. In addition, the suggestions ‎mentioned for further research should be well presented.‎

6. PLOS authors have the option to publish the peer review history of their article (what does this mean?). If published, this will include your full peer review and any attached files.

Reviewer #1: **Yes: **ZHOU ZHOU

Reviewer #2: No

Reviewer #3: No

---

## [Author Response · Author response to Decision Letter 0]

15 Mar 2024

The response to review comments have been attached in separate sheet.

---

## [Decision Letter · Decision Letter 1]

23 Apr 2024

E2SVM: Electricity-Efficient SLA-aware Virtual Machine Consolidation Approach in Cloud Data Centers

PONE-D-24-01820R1

Dear Dr. Aqeel,

We’re pleased to inform you that your manuscript has been judged scientifically suitable for publication and will be formally accepted for publication once it meets all outstanding technical requirements.

Kind regards,

Jacopo Soldani

Academic Editor

PLOS ONE

Additional Editor Comments (optional):

Reviewers' comments:

Reviewer's Responses to Questions

**Comments to the Author**

1. If the authors have adequately addressed your comments raised in a previous round of review and you feel that this manuscript is now acceptable for publication, you may indicate that here to bypass the “Comments to the Author” section, enter your conflict of interest statement in the “Confidential to Editor” section, and submit your "Accept" recommendation.

Reviewer #1: All comments have been addressed

Reviewer #2: All comments have been addressed

Reviewer #3: All comments have been addressed

2. Is the manuscript technically sound, and do the data support the conclusions?

Reviewer #1: Yes

Reviewer #2: Yes

Reviewer #3: Yes

3. Has the statistical analysis been performed appropriately and rigorously? 

Reviewer #1: Yes

Reviewer #2: Yes

Reviewer #3: Yes

4. Have the authors made all data underlying the findings in their manuscript fully available?

Reviewer #1: Yes

Reviewer #2: Yes

Reviewer #3: Yes

5. Is the manuscript presented in an intelligible fashion and written in standard English?

Reviewer #1: Yes

Reviewer #2: Yes

Reviewer #3: Yes

6. Review Comments to the Author

Reviewer #1: All of my concerns have been addressed (including the experiment part and other parts). It is suggested that you accept this paper.

Reviewer #2: (No Response)

Reviewer #3: (No Response)

7. PLOS authors have the option to publish the peer review history of their article (what does this mean?). If published, this will include your full peer review and any attached files.

Reviewer #1: **Yes: **Zhou Zhou

Reviewer #2: No

Reviewer #3: No

---

## [Editor Report · Acceptance letter]

4 May 2024

PONE-D-24-01820R1 

PLOS ONE

Dear Dr. Aqeel, 

I'm pleased to inform you that your manuscript has been deemed suitable for publication in PLOS ONE. Congratulations! Your manuscript is now being handed over to our production team.

Kind regards, 

on behalf of

Dr. Jacopo Soldani 

Academic Editor

PLOS ONE